# Microstructure and Superior Corrosion Resistance of an In-Situ Synthesized NiTi-Based Intermetallic Coating via Laser Melting Deposition

**DOI:** 10.3390/nano12040705

**Published:** 2022-02-20

**Authors:** Cheng Deng, Menglong Jiang, Di Wang, Yongqiang Yang, Vyacheslav Trofimov, Lianxi Hu, Changjun Han

**Affiliations:** 1School of Mechanical and Automotive Engineering, South China University of Technology, Guangzhou 510641, China; dengcheng@scut.edu.cn (C.D.); rong0610@mail.scut.edu.cn (M.J.); mewdlaser@scut.edu.cn (D.W.); meyqyang@scut.edu.cn (Y.Y.); trofimov@scut.edu.cn (V.T.); 2School of Materials Science and Engineering, Harbin Institute of Technology, Harbin 150001, China; hulx@hit.edu.cn

**Keywords:** laser melting deposition, nickel–titanium coating, in-situ synthesis, corrosion resistance

## Abstract

A nickel–titanium (NiTi)-based intermetallic coating was in-situ synthesized on a Ti–6Al–4V (TC4) substrate via laser melting deposition (LMD) using Ni–20Cr and TC4 powders. Scanning electron microscopy, X-ray diffraction, a digital microhardness tester and an electrochemical analyzer were used to evaluate the microstructure, Vicker’s microhardness and electrochemical corrosion resistance of the intermetallic coating. Results indicate that the microstructure of the intermetallic coating is composed of NiTi_2_, NiTi and Ni_3_Ti. The measured microhardness achieved is as high as ~850 HV_0.2_, ~2.5 times larger than that of the TC4 alloy, which can be attributed to the solid solution strengthening of Al and Cr, dispersion strengthening of the intermetallic compounds, and grain refinement strengthening from the rapid cooling of LMD. During the electrochemical corrosion of 3.5% NaCl solution, a large amount of Ti ions were released from the intermetallic coating surface and reacted with Cl^−^ ions to form [TiCl_6_]^2^ with an increase in corrosion voltage. In further hydrolysis reactions, TiO_2_ formation occurred when the ratio of [TiCl_6_]^2−^ reached a critical value. The in-situ synthesized intermetallic coating can achieve a superior corrosion resistance compared to that of the TC4 alloy.

## 1. Introduction

Nickel–titanium (NiTi) alloys are widely used in aerospace, electric, chemical, and biological medicine applications due to their shape memory effect, high strength, good wear resistance, pseudo-elasticity and biocompatibility [1,2,3,4]. In particular, NiTi alloys exhibit excellent corrosion resistance because of the formation of a stable and dense titanium oxide passivation film on their surface to prevent further corrosion [5,6,7]. However, the passivation film is susceptible to damage and even being detached from NiTi alloys in a harsh corrosion environment (e.g., highly acidified chloride solutions) [8,9,10,11].

NiTi-based intermetallic coatings have been developed to improve the corrosion resistance of NiTi alloys and reduce the material costs by decreasing the usage of expensive NiTi alloys. For instance, Zhou et al. prepared NiTi-based intermetallic coatings on a Cu substrate by low pressure plasma spraying (LPPS) [12]. They found that the coatings were composed of NiTi, NiTi_2_, Ni_3_Ti, and Ti, and possessed higher anti-cavitation performance than coatings of pure Ti. Bitzer et al. prepared NiTi-based intermetallic coatings by LPPS on a 42CrMo_4_ steel and found that the NiTi-based intermetallic coatings were composed of NiTi, oxygen-containing NiTi_2_ and Ni_4_Ti_3_ [13]. Such coatings showed significantly improved cavitation resistance compared with the UTP 730 stainless steel. However, defects, such as porosity and microcracks may deteriorate the corrosion resistance of LPPS-produced NiTi-based intermetallic coatings [14].

Laser surface modification techniques have been demonstrated to reduce the defects produced in NiTi-based coatings. Hiraga et al. produced dense NiTi-based intermetallic coatings using laser-plasma spraying hybrid cladding [14,15]. They found that the NiTi and Ni_3_Ti intermetallic compounds formed in the Ni_50_Ti_50_ coating, while NiTi, Ni_3_Ti and NiTi_2_ appeared in the Ni_60_Ti_40_ coating. As a result, the corrosion resistance of the NiTi-based intermetallic coatings was ~40 times higher than that of the TC4 alloy. Cui et al. [16] studied the corrosion resistance of NiTi-based intermetallic coatings produced by laser remelting and laser gas nitriding. The corrosion resistance of the coatings were enhanced by the formation of a TiN phase on the coating surface. Moreover, Hu et al. [10] added TaC particles in NiTi/NiTi_2_ composite coatings prepared by laser cladding for the enhancement of the corrosion resistance of the coatings. The TaC particles contributed to the formation of SiO_2_ and Ta_2_O_5_ thereby hindering the further corrosion of the composite coatings.

In this paper, we report a laser melting deposition (LMD) method of preparing the in-situ synthesized NiTi-based intermetallic coatings for surface modification. We consider LMD as an innovative and effective process for producing high-performance NiTi-based intermetallic coatings. LMD, characterized by a rapid cooling rate up to 10^6^–10^7^ K/s, is a new and promising laser surface treatment technique for strengthening pure metals, alloys, and metal matrix composites [17,18]; it has been shown to be effective in improving the mechanical and wear properties of a number of metals and alloys because of its capability to impart desirable refined microstructures and reinforced phases through rapid solidification and chemical reactions. Another advantage for laser deposition would be the achievement of very complex geometries and customized designs [19]. Although some published works that did report that fine microstructure and superior wear resistance of NiTi-based intermetallic alloy coatings in-situ synthesized through LMD can be achieved [1,20], surprisingly few have reported on the electrochemical corrosion behavior of the in-situ synthesized NiTi-based intermetallic coatings.

In this work, Ni–20Cr and TC4 powders were utilized and mixed as the cladding materials to reduce the cost of raw powders. The microstructure and mechanics of a NiTi-based intermetallic coating in-situ synthesized by optimized LMD process were characterized using scanning electron microscopy, X-ray diffraction, and digital microhardness tester. Furthermore, for the first time, the corrosion behavior of the intermetallic coating was evaluated by electrochemical corrosion and immersion tests, and the underlying mechanism of the enhanced corrosion resistance of the in-situ synthesized NiTi-based coating was discussed.

## 2. Materials and Methods

### 2.1. Materials

A TC4 alloy with dimensions of 100 × 60 × 10 mm^3^ was used as the substrate for the coating. The chemical composition of the TC4 alloy is listed in Table 1. Ni–20Cr (wt%) and TC4 powders with a weight ratio of 4:1 were mechanically mixed at a speed of 300 rpm for 1 h in an alcohol atmosphere in a planetary ball mill (TJ-2L, procured from a company in Tianjin, China, TECHIN Ltd.). The powder-to-ball mass ratio was set as 1:3. The mixed powder was dried at 373 K for 2 h and then used as the cladding material.

### 2.2. LMD Coating Process

Prior to LMD, the surface of the TC4 substrate was polished and cleaned with alcohol. The LMD process was carried out using an IPG fiber laser with a wavelength of 1070 nm. A shielding of argon gas was used to protect the molten pool of LMD and deliver the mixed powders into the molten pool. A schematic configuration of LMD is depicted in Figure 1. The processing parameters were optimized to obtain the crack-free NiTi-based intermetallic coating by LMD: the laser power of 1 kW, the laser scanning speed of 600 mm/min, powder feeding rate of 0.8 g/min, the spot diameter of 2 mm, and overlapping rate of 50%. The height and width of NiTi-based intermetallic coating were ~1 mm and ~7 mm, respectively.

After LMD, all the samples were cut from the substrate, polished, and then etched by Kroll’s solution (10 mL HF, 15 mL HNO_3_, and 75 mL H_2_O). The microstructure of the NiTi-based intermetallic coating prepared by LMD was examined by field emission scanning electron microscopy (SEM, ZEISS Sigma 300) equipped with an X-ray energy-dispersive spectrometer (EDS). The phases were investigated by a D/MAX-2500 X-ray diffraction (XRD, Cu Kα at 40 kV and 40 mA, scanning rate of 0.02°/s). The microhardness of the intermetallic coating was tested using a HV-1000 Vickers digital microhardness tester with a load of 1.96 N and a dwelling time of 10 s. The reported microhardness was averaged from three samples for each condition.

The electrochemical corrosion resistance of the NiTi-based intermetallic coating was measured by a CHI604E electrochemical analyzer (Chenhua, Shanghai, China) in a 3.5 wt% NaCl solution. A standard three-electrode cell was composed of a working electrode made from a composite specimen with an exposed area of 1 cm^2^, a platinum counter electrode, and a saturated calomel reference electrode. All the samples were immersed into the 3.5 wt% NaCl solution at room temperature for 1 h to stabilize the open circuit potential (OCP). Potentiodynamic polarization scanning was varied from −1.5 V to 4.0 V at a sweep rate of 5 mV/s. Electrochemical impedance spectroscopy (EIS) testing was performed at the OCP potentionstatically by scanning a frequency range from 10^−2^~10^5^ Hz with a voltage perturbation amplitude of 10 mV. The corresponding Nyquist and Bode plots were fitted by impedance spectrum data using Zsimpwin software. All potentials were measured at least three times.

Prior to immersion testing, the samples were ground with waterproof silicon carbide papers up to 2000 grits under running water, then cleaned in acetone, ethanol for 30 min using ultrasound, and finally dried at room temperature. The static immersion testing was conducted using 3.5% NaCl solution for 7 days at room temperature. Three samples were tested for each group.

## 3. Results and Discussion

### 3.1. Microstructure of the NiTi-Based Intermetallic Coating

Figure 2a shows the XRD pattern of the NiTi-based intermetallic coating in-situ synthesized by LMD. The results indicate that it consists of a dominant NiTi_2_ phase with a face-centered cubic (fcc) structure, while the other two intermetallic phases of NiTi and Ni_3_Ti have primitive hexagonal crystal structures (Figure 2a). The grain sizes of each phase can be determined from the Bragg peak width at half of the maximum intensity using the Scherrer formula [21]:(1)$$D=\frac{0.9\lambda }{\mathrm{Bcos}\theta },$$
where D is the grain size, λ is the wavelength of the X-ray radiation, B is the peak width at half of the maximum intensity, and θ is the Bragg diffraction angle. The grain sizes of NiTi, NiTi_2_ and Ni_3_Ti were calculated as ~30 nm, ~17 nm and ~25 nm, respectively. According to the previous works [22,23], the volume fractions of the NiTi, NiTi_2_ and Ni_3_Ti phases can be calculated as ~20%, ~52% and ~28%, respectively, (as shown in Figure 2b).

Figure 3 shows the microstructure of the NiTi-based intermetallic coating in-situ synthesized by LMD. As shown in Figure 3a, the presence of three distinct regions (as marked by I, II and III) can be observed from the top to the bottom of the coating. The petal-like dendritic microstructure is formed in region I, with a grain size of a maximum of ~20 µm in length (marked as A). Based on the XRD results (Figure 2a) and EDS analysis (Table 2), the petal-like dendrites can be identified as NiTi, while the intermetallic (marked B) located between the petal-like dendrites is identified as Ni_3_Ti. Moreover, region II is mainly composed of equiaxial or columnar dendrites (marked as C and D), which can be identified as NiTi_2_. The length of the oriented dendrites is measured as ~80 µm and the main stems are angled ~30° toward the normal direction of the coating-substrate interface (marked as D). 

Generally, thermocapillarity caused the violent stirring and convection in the molten pool [24], thus, increasing the longer lifespan in the center of the molten pool than that at the bottom of the molten pool; this, in turn, resulted in the non-uniform distribution of solutes and the temperature in front of the solid/liquid interface. Therefore, the growth direction of dendrites can deviate from the normal direction of the coating/substrate interface. However, coarse dendritic arms can be formed because of a relatively slow solidification speed at the bottom of the molten pool [25]. It is seen that large secondary dendritic arms appear at the bottom of region II (marked E), and the lateral growths of dendrites occur near the coating/substrate interface (marked F); these phenomena are attributed to a relatively larger specific surface area of the smaller dendritic arms, facilitating the growths of larger dendritic arms by way of consuming the smaller dendritic arms to reduce the total surface energy. The longer time the dendrites coarsening takes, the larger spacing the dendritic arms possess. Hence, an increase in the distance from the coating/substrate interface can decrease the spacing of secondary dendritic arms.

In addition, planar growths are seen at region III; such features indicate that the metallurgical bonding is generated at the coating/substrate interface. According to rapid solidification theory, the characteristics of the microstructure growths are related to the ratio G/R, where G is the temperature gradient and R is the solidification front rate. The R value is related to the laser scanning speed V_S_ directly and can be described as follows [26,27]:(2)$$R=V_{s}\cos\theta$$
where θ is the angle between V_S_ and R, h is the cladding height, and A is the spot diameter, as schematized in Figure 4. Prominently, R starts off with zero at the bottom of the molten pool but increases rapidly to the maximum value. However, G starts off with the largest value at the bottom of the molten pool while decreasing gradually toward the surface of the molten pool. Therefore, the G/R value approaches an infinite value at the bottom of the molten pool just corresponding to the planar growths. With the increasing distance far away from the coating/substrate interface, the G/R value decreases, inferring the presence of a constitutional supercooling ahead of the solidification front. Hence, the planar solid/liquid interface becomes unstable, resulting in the formation of dendrites (Figure 3d).

During LMD, there was a rapid phase transformation from the liquid phase to β-Ti for the cladding powders. According to the phase diagram of Ni-Ti alloy (Figure 5) [28], when the thermal diffusion continued, Ni_3_Ti and NiTi formed with a eutectic reaction occurring in the liquid phase at a temperature of 1583 K.
(3)$$L\to \mathrm{NiTi}+{\mathrm{Ni}}_{3}\mathrm{Ti}$$

As the atom ratio of Ni to Ti is 77.99:86.33 (less than 1:1) in this work, a peritectic reaction between the liquid phase L′ and the formed NiTi in the titanium-rich side can proceed, resulting in the formation of another intermetallic compound NiTi_2_ at a temperature of 1257 K [28,29,30]:(4)$$L\prime +\mathrm{NiTi}\to {\mathrm{NiTi}}_{2}$$

For Ni-Ti binary system at different temperatures, NiTi (formation enthalpy ∆*H* = −67 kJ/mol), NiTi_2_ (∆*H* = −83 kJ/mol), Ni_3_Ti (∆*H* = −140 kJ/mol) intermetallic compounds can be formed with exothermic reactions occurring [31,32,33]. Ni_3_Ti can be formed firstly during LMD because of its minimum formation enthalpy. According to Equation (4), when cooling proceeds, NiTi_2_ can be produced from the interaction between the formed NiTi and the residual liquid phase L′. This is the reason to explain the presence of dominant NiTi_2_ phase constituent in the intermetallic coating (Figure 2).

The formation mechanism of the petal-like dendrites can be explained as follows. The intermetallic compounds (NiTi and Ni_3_Ti) can grow rapidly at the initial stage of rapid solidification due to constitutional supercooling. Afterward, the Ni_3_Ti grows rapidly into the coarse dendritic branch and the Ti atoms diffuse into the liquid phase. The supersaturated Ni-based phase containing rich Ti atoms is precipitated, as the subsequent phase transforms into a NiTi intermetallic and grows on the surface of the primary Ni_3_Ti intermetallic. Consequently, the duplex phase nucleation sites with the interface of the intergrowth are formed, supplying the atoms for the neighboring phase to grow harmoniously, which depends on the diffusivity of the solute atoms, such as Ni, Cr and Ti to diffuse continually on the interface of Ni_3_Ti and NiTi intermetallics. Moreover, the eutectic Ni_3_Ti and NiTi phases are characterized by the non-facet growth of the unshaped interface. As such, the eutectics are formed by the intergrowth of Ni_3_Ti and NiTi, both of which present different crystal structures (Figure 3b). Therefore, the petal-like eutectic intermetallics grow in terms of the intergrowth model of layer and slice during LMD.

### 3.2. Microhardness of NiTi-Based Intermetallic Coating 

Figure 6 shows the microhardness of the NiTi-based intermetallic coating in-situ synthesized by LMD. It was observed that the average microhardness of the NiTi-based intermetallic coating is ~850 HV_0.2_, which is ~2.5 times that of the substrate (~350 HV_0.2_); this can be attributed to the formation of NiTi_2_, dispersion strengthening, solid solution strengthening, and grain refinement strengthening. First, the very important factor is the presence of the dominant NiTi_2_ phase in a face-centered cubic (fcc) structure with high hardness (HV700) and strong atomic bonds, thereby, increasing the overall hardness of NiTi-based intermetallic coating. In addition, these intermetallic compounds, such as NiTi and Ni_3_Ti (Figure 2) derived from the in-situ reactions of Ni and Ti atoms during LMD of Ni-20Cr and TC4 powders, are dispersedly distributed in the NiTi-based intermetallic coating, creating the dispersion strengthening effect. Moreover, the diffusion of a large number of alloying elements, such as Al and Cr into the NiTi, NiTi2 and Ni3Ti, results in their lattice distortions and solid solution strengthening. Finally, the formation of fine-grained dendrites in the intermetallic coating due to rapid solidification is essential to increase the overall hardness as well.

### 3.3. Electrochemical Corrosion of the NiTi-Based Intermetallic Coating

Figure 7 shows the anodic polarization curves of the NiTi-based intermetallic coating and Ti6Al4V alloy in 3.5% NaCl solution at room temperature. A distinct passivation behavior can be observed between the intermetallic coating and TC4 alloy, and the passivation region of the TC4 alloy is significantly larger than that of the NiTi-based intermetallic coating. The presence of a stable passivation platform for the TC4 alloy initiates from the corrosion voltage reaching around −0.3 V. However, the stable passivation of the intermetallic coating is formed at a corrosion voltage beyond 3 V, and a successive fluctuation of the curve in the range of −0.3 V to 3 V can be observed. This fluctuation can be attributed to the formation of different passivation films on the intermetallic coating surface, which is derived from different distributions of the intermetallic phases, such as NiTi_2_, NiTi and Ni_3_Ti. As a result, the polarization curve is changed from a passivation state to an active state, resulting in the instability of the passivation platform [34]. 

The corrosion potential *E*_corr_, corrosion current density *I*_corr_, and passivation current density *I*_p_ are the important parameters to evaluate the corrosion resistance of materials, as shown in Table 3. The *E*_corr_ value of the NiTi-based intermetallic coating is higher than that of the TC4 alloy, indicating much better stability of the passivation film formed in the coating [35,36]. The intermetallic coating obtains the *I*_corr_ value of 1.977 × 10^−7^ A/cm^2^, which is slightly smaller than that of the TC4 alloy (2.068 × 10^−7^ A/cm^2^). A lower *I*_corr_ indicates a smaller corrosion rate of the passivation film and better corrosion resistance [37,38]. In addition, when the corrosion voltage is less than 1 V, the *I*_p_ value of the intermetallic coating is smaller than that of the TC4 alloy. A larger *I*_p_ results in the faster dissolution of the passivation film. Therefore, the intermetallic coating is beneficial for improving the corrosion resistance of the TC4 alloy substrate. 

To further study the corrosion characteristics of the intermetallic coating and TC4 alloy, electrochemical impedance spectroscopy (EIS) was measured in a 3.5% NaCl solution. The Nyquist results of EIS are illustrated in Figure 8a. It is seen that a similar semicircle capacitive impedance loop exists in the Nyquist curves of both the intermetallic coating and TC4 alloy. However, the radii of the capacitive impedance loop of the intermetallic coating are larger than that of the TC4 alloy. The larger the radius of the capacitive impedance loop indicates the higher corrosion resistance [39,40]. Figure 8b,c show the Bode results of EIS. The maximum |Z| value of the intermetallic coating and TC4 alloy present a linear change with a slope of ~−1 in the low-frequency region (10^−2^–10^0^ Hz) and the intermediate frequency region (10^−2^~10^3^ Hz). 

The phase angles of the intermetallic coating and TC4 alloy increase, and the phase angle of the intermetallic coating (~90°) is slightly higher than that of TC4 alloy, indicating a typical capacitance behavior. The greater angle values of the coating indicate its better corrosion resistance, which is consistent with the results of polarization curves (Figure 7). This can be confirmed by the equivalent circuits shown in Figure 8d. To obtain the optimal fitting results, the equivalent circuits chosen χ^2^ (chi-squared) must be in the range of 10^−4^–10^−3^ and the results are listed in Table 4. As shown in Figure 8d, R_s_ can be regarded as the solution resistance, R_t_ is the resistance of the passivation film on the TC4 alloy surface during electrochemical corrosion, and R_ct_ is defined as the charge transfer resistance. The values of R_s_ and R_t_ of the intermetallic coating are smaller than those of the TC4 alloy, while the R_ct_ value of the intermetallic coating is much larger than that of the TC4 alloy. The higher R_ct_ indicates the higher charge transfer resistance and resultant better corrosion resistance. 

The equivalent circuit of the intermetallic coating is mainly composed of the double layer capacitance and constant phase angle. This indicates that double layer capacitance C_dl_ consists of the NaCl solution and coating, and the constant phase angle (CPE) is composed of the coating and TC4 substrate. Generally, CPE is defined as Z_CPE_ = [Z_0_(jw)^n^]^−1^, where Z_0_ is the constant of CPE, j^2^ = −1 is imaginary, w is the angular frequency (w = 2πf), and n is the index of CPE (−1 ≤ n ≤ 1) [41,42,43]. As shown in Table 4, the n value of the intermetallic coating (0.8628) is slightly greater than that of the TC4 alloy (0.8548), indicating that the passivation film on the intermetallic coating is denser than that on the TC4 alloy. Furthermore, the χ^2^ (chi-squared) values of the intermetallic coating and the TC4 alloy are all in the order of ~10^−4^, showing a good fitting result.

During electrochemical corrosion, the different concentrations of Cl^−^ ions agglomerate together on the surface of the intermetallic coating to replace the internal O ions, resulting in pitting. The schematic of the pitting formation process is illustrated in Figure 9. The corrosion current density increases quickly following the pitting process. With an increase in the corrosion voltage, the passivation platform of the intermetallic coating is punctured and a large amount of Ti ions are released to react with Cl^−^ ions and form [TiCl_6_]^2−^. When the corrosion voltage is increased to ~3 V, [TiCl_6_]^2−^ reaches a certain critical value in the solution, and hereby, TiO_2_ is produced from the hydrolysis reaction to protect the intermetallic coating from further corrosion [5,44]. Moreover, slight pitting can be found on the surface of the intermetallic coating (Figure 10a), while it becomes serious on the surface of the TC4 alloy (Figure 10b). This confirms that the intermetallic coating has superior corrosion resistance than the TC4 alloy during the electrochemical corrosion of 3.5% NaCl solution.

The intermetallic coating and TC4 alloy were immersed into the 3.5 wt% NaCl solution for 7 days to validate the results of polarization curves and EIS. Figure 11 shows the surface morphology of the intermetallic coating and TC4 alloy after the immersion testing. The surface of the intermetallic coating is relatively smooth and only slight pitting can be observed on the surface of the intermetallic coating (Figure 11a). Comparatively, the sizes and amounts of the pits on the TC4 surface are much larger than those on the surface of the intermetallic coating (Figure 11b). Therefore, the immersion results also demonstrate better corrosion resistance for the intermetallic coating than that of the TC4 alloy, which is in good agreement with the results of polarization curves and EIS.

## 4. Conclusions

This work investigates the microstructure, mechanical properties, and electrochemical corrosion resistance of a NiTi-based intermetallic coating in-situ synthesized by LMD. The main findings are presented as follows.

(1)The NiTi-based intermetallic coating was in-situ synthesized on the TC4 substrate by LMD using a mixed powder of Ni-20Cr and TC4. The phases of the coating are composed of the intermetallic compounds of NiTi_2_, NiTi, and Ni_3_Ti, and their volume fractions are ~52%, ~20% and ~28%, respectively.(2)The microhardness of the intermetallic coating is ~850 HV_0.2_, which is ~2.5 times larger than that of the TC4 alloy. The high microhardness can be attributed to the solid solution strengthening of Al and Cr, dispersion strengthening of the intermetallic compounds, and grain refinement strengthening from the rapid solidification.(3)The intermetallic coating exhibits better corrosion resistance than the TC4 alloy. With the increase in the corrosion voltage, a large amount of Ti ions react with the Cl^−^ ions to form [TiCl_6_]^2−^ in the solution. When [TiCl_6_]^2−^ reaches a certain critical value, TiO_2_ is formed by hydrolysis reaction to protect the intermetallic coating from further corrosion. Slight pitting appears on the coating surface, while large pits can be observed on the TC4 surface.

## Figures and Tables

**Figure 1 nanomaterials-12-00705-f001:**
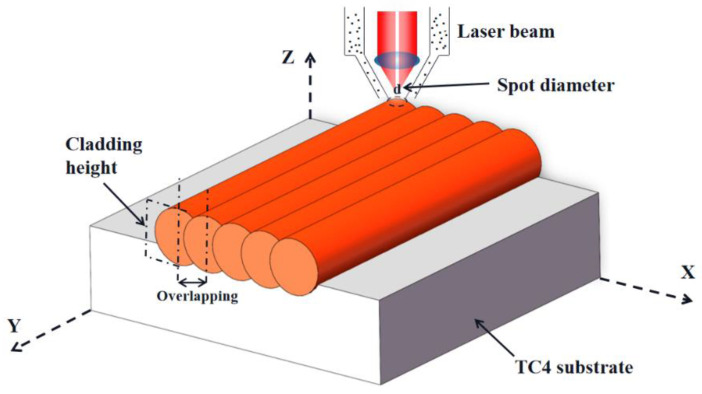
Schematic drawing of laser melting deposition.

**Figure 2 nanomaterials-12-00705-f002:**
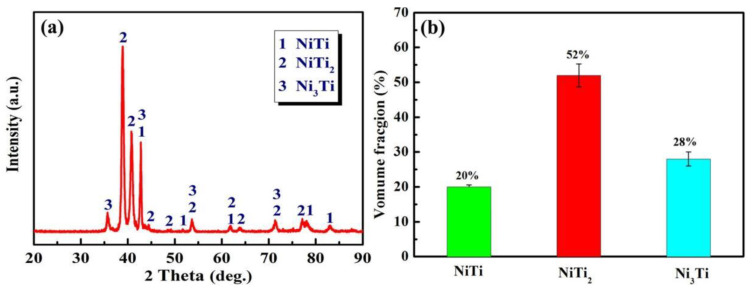
(**a**) X-ray diffraction (XRD) pattern; (**b**) The estimated volume fraction of phase constituents in the intermetallic coating.

**Figure 3 nanomaterials-12-00705-f003:**
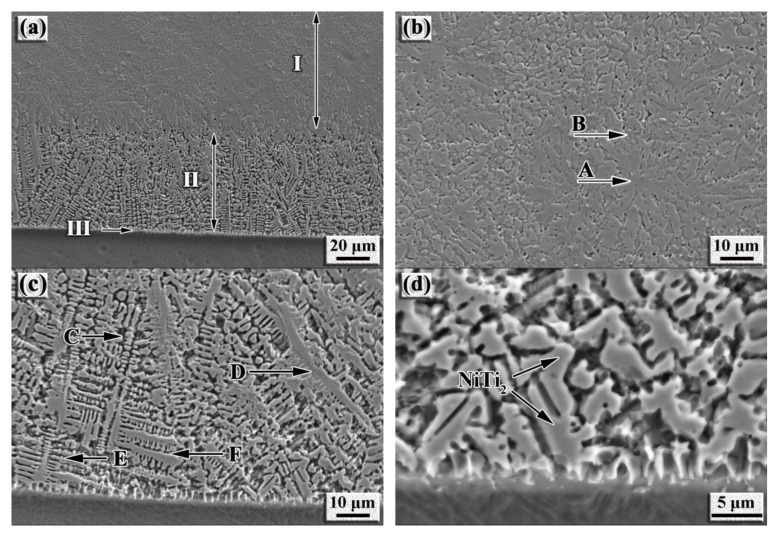
SEM images showing the microstructure of the in-situ synthesized NiTi-based intermetallic coating via LMD: (**a**) Cross-sectional morphology; (**b**) Region I; (**c**) Region II; (**d**) Region III.

**Figure 4 nanomaterials-12-00705-f004:**
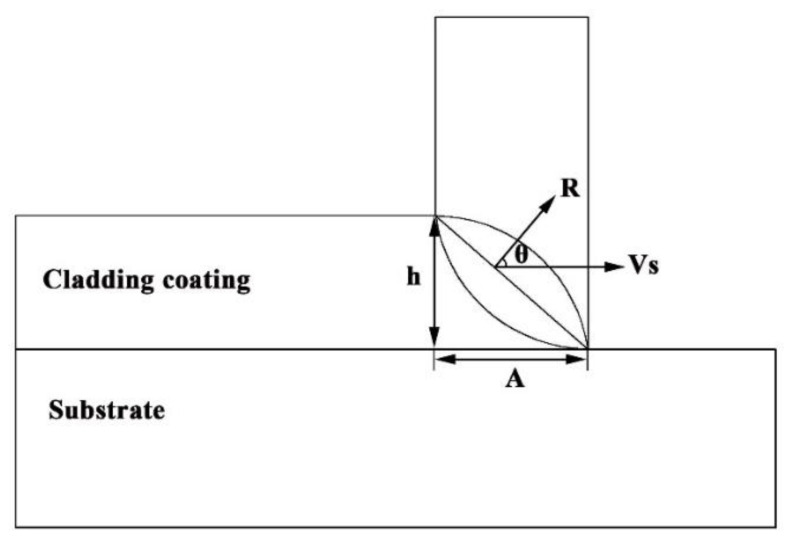
Schematic diagram of angle relationship between the solidification front rate and the laser scanning speed.

**Figure 5 nanomaterials-12-00705-f005:**
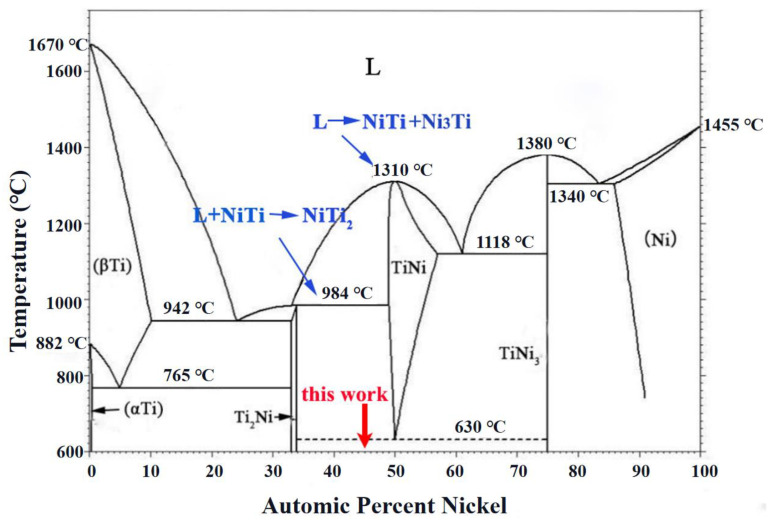
Phase diagram of Ni-Ti alloy showing phase transformation at different temperatures [28].

**Figure 6 nanomaterials-12-00705-f006:**
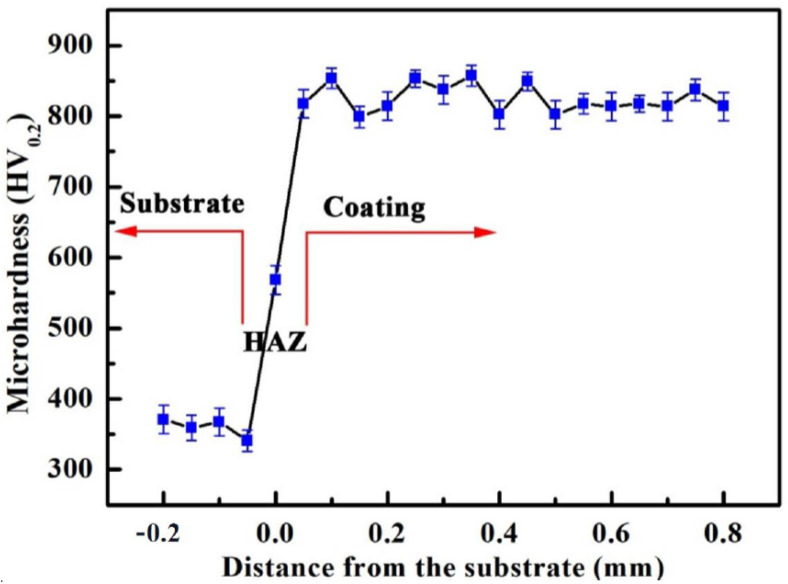
Microhardness of the NiTi-based intermetallic coating. HAZ represents heat affected zone.

**Figure 7 nanomaterials-12-00705-f007:**
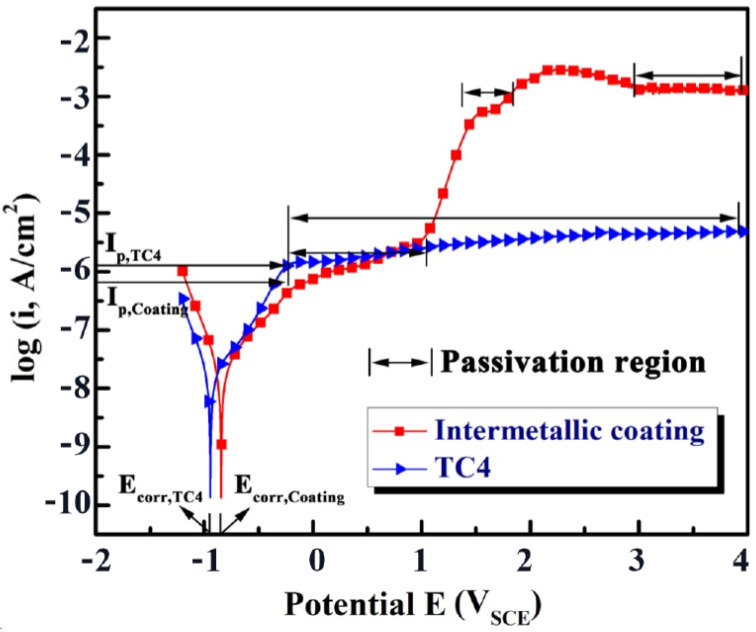
Potential dynamic curves of the NiTi-based intermetallic coating and TC4 alloy in the 3.5 wt% NaCl solution.

**Figure 8 nanomaterials-12-00705-f008:**
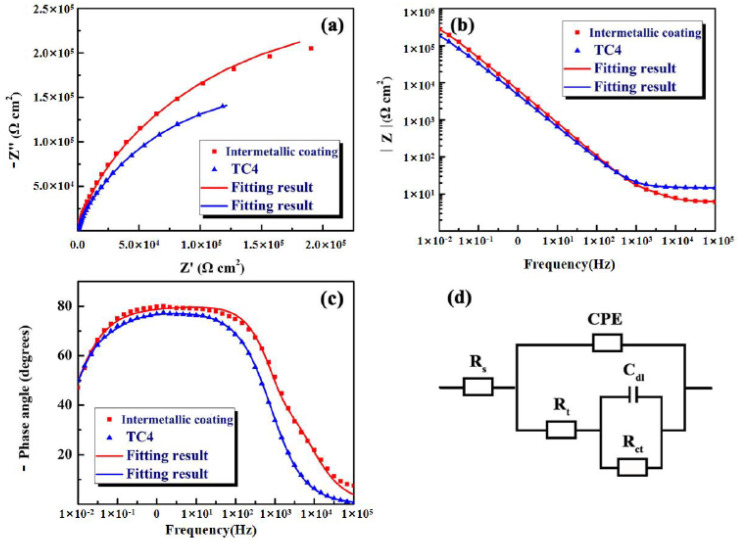
EIS results of the NiTi-based intermetallic composite coating and TC4 alloy in the 3.5 wt% NaCl solution: (**a**) Nyquist curves; (**b**,**c**) Bode curves; (**d**) Equivalent circuit.

**Figure 9 nanomaterials-12-00705-f009:**
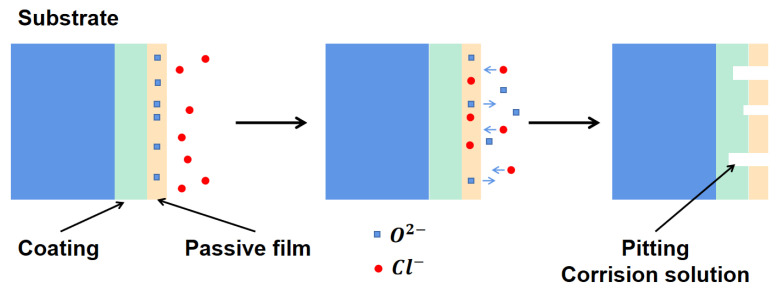
Schematic illustration of the pitting of the NiTi-based intermetallic coating in the 3.5 wt% NaCl solution.

**Figure 10 nanomaterials-12-00705-f010:**
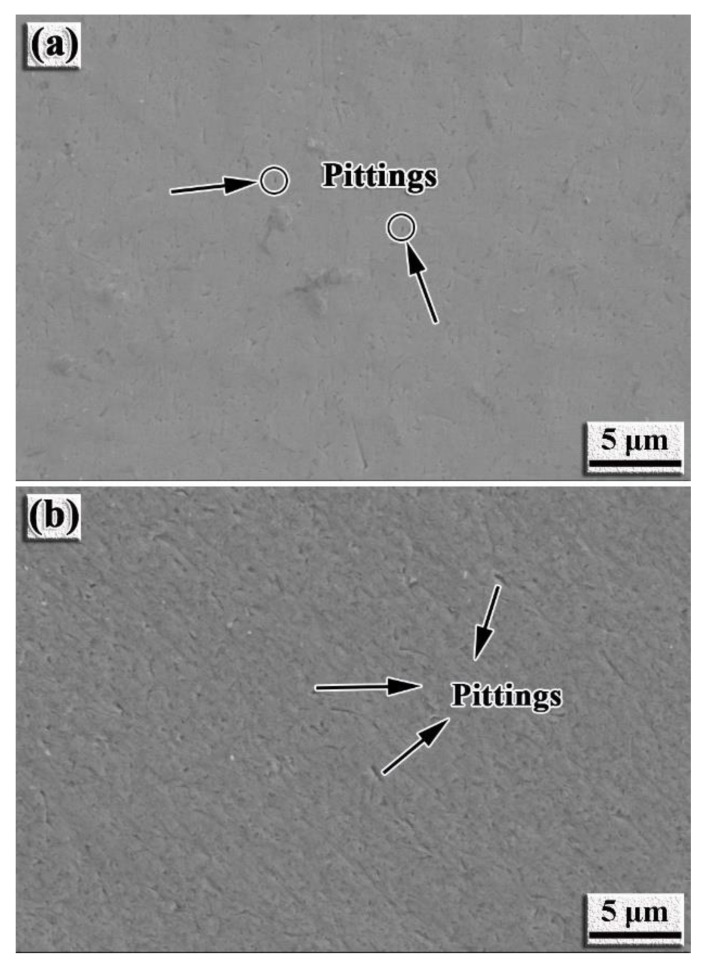
SEM images showing the morphology of the samples after electrochemical corrosion: (**a**) NiTi-based intermetallic coating; (**b**) TC4 alloy.

**Figure 11 nanomaterials-12-00705-f011:**
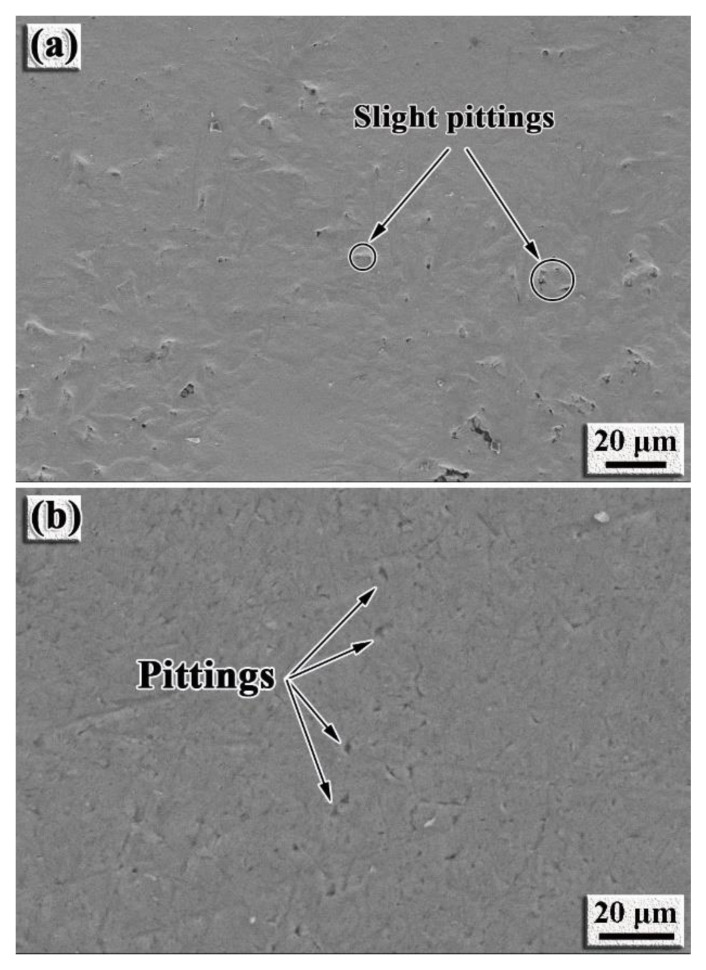
SEM images showing the morphology of the samples after immersing test in the 3.5 wt.% NaCl solution for 7 days at room temperature: (**a**) NiTi-based intermetallic coating; (**b**) TC4 alloy.

**Table 1 nanomaterials-12-00705-t001:** Chemical composition of TC4 powder used in this work.

Element	C	H	O	N	Fe	Al	V	Ti
Composition (wt%)	0.08	0.015	0.2	0.05	0.4	6.05	4.02	Bal.

**Table 2 nanomaterials-12-00705-t002:** Element compositions of the intermetallic coating from EDS measurement.

Location	Composition (wt%)				
	Al	Ti	V	Cr	Ni
A	2.98	44.33	1.63	8.01	43.01
B	3.93	56.61	1.87	4.93	32.66
C	3.59	56.79	1.61	6.86	31.16
D	4.1	47.27	4.27	16.95	27.41
E	1.84	58.99	1.48	3.42	34.27
F	5.14	56.86	1.49	3.89	32.62

**Table 3 nanomaterials-12-00705-t003:** Corrosion parameters of the intermetallic coating and TC4 alloy in Figure 7.

Sample	E_corr_ (V)	I_corr_ (A/cm^2^)	I_p_ (A/cm^2^)
Immiscible coating	−0.854	1.977 × 10^−7^	−6.2 ± 0.01
TC4 alloy	−0.943	2.068 × 10^−7^	−5.8 ± 0.01

**Table 4 nanomaterials-12-00705-t004:** Electrochemical results obtained from equivalent circuits fitting of the intermetallic coating and TC4 alloy in the 3.5% NaCl solution.

Sample	R_S_ (Ω cm^2^)	R_t_ (Ω cm^2^)	R_ct_ (Ω cm^2^)	C_d1_ (F cm^−2^)	Q1-Y_0_ (Ω^−^^1^ cm^2^s^n^)	n_1_	χ^2^
NiTi-based coating	6.931	23.99	5.69 × 10^5^	4.94 × 10^−6^	2.63 × 10^−5^	0.8628	8.6 × 10^−4^
TC4 alloy	14.82	2.459 × 10^5^	1.33 × 10^5^	3.27 × 10^−5^	4.29 × 10^−5^	0.8548	1.2 × 10^−4^

## Data Availability

Not applicable.

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
