# Peer review of "Microstructure and Superior Corrosion Resistance of an In-Situ Synthesized NiTi-Based Intermetallic Coating via Laser Melting Deposition"

_nanomaterials, 2022, doi:10.3390/nano12040705_

Round 1

Reviewer 1 Report

According to the manuscript with title: “Microstructure and superior corrosion resistance of an in-situ synthesized NiTi-based intermetallic coating via laser melting deposition". The submitted work is introducing a new valuable and interesting idea and the given results confirm the idea. This work is suitable for publication in the Journal. I suggest the acceptance after some corrections as follows;

  1. Add critical results in abstract section
  2. Give some results with numbers in conclusion
  3. Add more explanation to experimental work
  4. Reformulate the novelty of the work in introduction
  5. The bibliography needs to be improved. Some papers in literature could be taken into consideration such as; The role of Li4Ti5O12 nanoparticles on enhancement the performance of PVDF/PVK blend for lithium-ion batteries; Chitosan/graphene oxide composite as an effective removal of Ni, Cu, As, Cd and Pb from wastewater; Laser-assisted for preparation ZnO/CdO thin film prepared by pulsed laser deposition for catalytic degradation; Wound dressing properties of functionalized environmentally biopolymer loaded with selenium nanoparticles; Differentiation between cellulose acetate and polyvinyl alcohol nanofibrous scaffolds containing magnetite nanoparticles/graphene oxide via pulsed laser ablation technique for tissue engineering application; Differentiation between cellulose acetate and polyvinyl alcohol nanofibrous scaffolds containing magnetite nanoparticles/graphene oxide via pulsed laser ablation technique for tissue engineering applications; Precipitation of silver nanoparticle within silicate glassy matrix via Nd:YAG laser for biomedical applications; Casted polymeric blends of Carboxymethyl cellulose/polyvinyl alcohol doped with gold nanoparticles via pulsed laser ablation technique; morphological features, optical and electrical investigation. --It is good to mention and add all these articles that could be important in the introduction section and add to references section
  6. Correct typographical errors.
  7. Don’t use abbreviations in title and abstract, you must define it in first time use

Author Response

1.    Add critical results in abstract section
We add all critical results in abstract section
2.    Give some results with numbers in conclusion
We listed numbers for results in conclusion
3.    Add more explanation to experimental work
We add the schematic for laser melting deposition process for better explanation
4.    Reformulate the novelty of the work in introduction
We rewrite the novelty and add Reviewer 1 recommendation reference paper in [19]: Laser-assisted for preparation ZnO/CdO thin film prepared by pulsed laser deposition for catalytic degradation
5.    The bibliography needs to be improved. Some papers in literature could be taken into consideration such as; The role of Li4Ti5O12 nanoparticles on enhancement the performance of PVDF/PVK blend for lithium-ion batteries; Chitosan/graphene oxide composite as an effective removal of Ni, Cu, As, Cd and Pb from wastewater; Wound dressing properties of functionalized environmentally biopolymer loaded with selenium nanoparticles; Differentiation between cellulose acetate and polyvinyl alcohol nanofibrous scaffolds containing magnetite nanoparticles/graphene oxide via pulsed laser ablation technique for tissue engineering application; Differentiation between cellulose acetate and polyvinyl alcohol nanofibrous scaffolds containing magnetite nanoparticles/graphene oxide via pulsed laser ablation technique for tissue engineering applications; Precipitation of silver nanoparticle within silicate glassy matrix via Nd:YAG laser for biomedical applications; Casted polymeric blends of Carboxymethyl cellulose/polyvinyl alcohol doped with gold nanoparticles via pulsed laser ablation technique; morphological features, optical and electrical investigation. --It is good to mention and add all these articles that could be important in the introduction section and add to references section
We add Reviewer 1 recommendation reference paper in [19]: Laser-assisted for preparation ZnO/CdO thin film prepared by pulsed laser deposition for catalytic degradation
6.    Correct typographical errors.
We corrected this.
7.    Don’t use abbreviations in title and abstract, you must define it in first time use
We give the abbreviations definition for all in first time use.

Reviewer 2 Report

Authors present a very good paper. The introduction section is well written with a clearly stated goal. Experimental is sufficient, so other riders can repeat the experiments. The R&D section is clear without unnecessary details. Literature is up to date, so the reviewer suggests acceptance as is.

Author Response

Thanks for Reviewer 2. 

Reviewer 3 Report

"Figure 1d" - row 177 - I think it needs to be reviewed

I recommend revising / reformulating the text from the rows 238, 239 and first part of 240.

Author Response

1.    "Figure 1d" - row 177 - I think it needs to be reviewed It is Fig 2d
We fixed this, after insert a new figure for the schematic of LMD process, we corrected "Figure 1d" - row 177 as Figure 3d.

2.    I recommend revising / reformulating the text from the rows 238, 239 and first part of 240.
We rewrite this sentence as: “However, the stable passivation of the intermetallic coating is formed at the corrosion voltage beyond 3V, and a successive fluctuation of the curve in the range of ‒0.3V to 3V can be observed.”